# Characterization and Performance of Peanut Shells in Caffeine and Triclosan Removal in Batch and Fixed-Bed Column Tests

**DOI:** 10.3390/molecules29122923

**Published:** 2024-06-19

**Authors:** Cristina E. Almeida-Naranjo, Mayra Frutos, Victor H. Guerrero, Cristina Villamar-Ayala

**Affiliations:** 1Grupo de Biodiversidad Medio Ambiente y Salud (BIOMAS), Facultad de Ingeniería y Ciencias Aplicadas, Universidad de Las Américas, Redondel del Ciclista Antigua Vía a Nayón, Quito 170124, Ecuador; 2Department of Civil and Environmental Engineering, Escuela Politécnica Nacional, Ladrón de Guevara E1-253, Quito 170525, Ecuador; mayralejandrafrutos@gmail.com; 3Department of Materials, Escuela Politécnica Nacional, Ladrón de Guevara E11-253, Quito 170525, Ecuador; victor.guerrero@epn.edu.ec; 4Departamento de Ingeniería en Obras Civiles, Facultad de Ingeniería, Universidad Santiago de Chile (USACH), Av. Victor Jara 3659, Estación Central, Santiago 9170022, Chile; cristina.villamar@usach.cl; 5Programa Para el Desarrollo de Sistemas Productivos Sostenibles, Facultad de Ingeniería, Universidad de Santiago de Chile (USACH), Av. Victor Jara 3769, Estación Central, Santiago 9170022, Chile

**Keywords:** emerging contaminant, sustainable adsorption materials, versatility in wastewater treatment, adsorption efficiency, particle size effect

## Abstract

Peanut shells’ adsorption performance in caffeine and triclosan removal was studied. Peanut shells were analyzed for their chemical composition, morphology, and surface functional groups. Batch adsorption and fixed-bed column experiments were carried out with solutions containing 30 mg/L of caffeine and triclosan. The parameters examined included peanut shell particle size (120–150, 300–600, and 800–2000 µm), adsorbent dose (0.02–60 g/L), contact time (up to 180 min), bed height (4–8 cm), and hydraulic loading rate (2.0 and 4.0 m^3^/m^2^-day). After determining the optimal adsorption conditions, kinetics, isotherm, and breakthrough curve models were applied to analyze the experimental data. Peanut shells showed an irregular surface and consisted mainly of polysaccharides (around 70% lignin, cellulose, and hemicellulose), with a specific surface area of 1.7 m^2^/g and a pore volume of 0.005 cm^3^/g. The highest removal efficiencies for caffeine (85.6 ± 1.4%) and triclosan (89.3 ± 1.5%) were achieved using the smallest particles and 10.0 and 0.1 g/L doses over 180 and 45 min, respectively. Triclosan showed easier removal compared to caffeine due to its higher lipophilic character. The pseudo-second-order kinetics model provided the best fit with the experimental data, suggesting a chemisorption process between caffeine/triclosan and the adsorbent. Equilibrium data were well-described by the Sips model, with maximum adsorption capacities of 3.3 mg/g and 289.3 mg/g for caffeine and triclosan, respectively. In fixed-bed column adsorption tests, particle size significantly influenced efficiency and hydraulic behavior, with 120–150 µm particles exhibiting the highest adsorption capacity for caffeine (0.72 mg/g) and triclosan (143.44 mg/g), albeit with clogging issues. The experimental data also showed good agreement with the Bohart–Adams, Thomas, and Yoon–Nelson models. Therefore, the findings of this study highlight not only the effective capability of peanut shells to remove caffeine and triclosan but also their versatility as a promising option for water treatment and sanitation applications in different contexts.

## 1. Introduction

The extensive use of chemical products such as pesticides, hormones, pharmaceuticals, and personal care items amounts to approximately 2.0 million tons per year [1]. Their waste/byproducts are known as emerging contaminants (ECs). Despite their typically low concentrations, ranging from nanograms per liter to micrograms per liter, they have led to widespread water contamination [2]. ECs exhibit different behaviors in the environment due to their distinct physicochemical properties. Some of them are toxic (e.g., K_ow_ bisphenol A = 2.2–3.4), persistent (e.g., pesticides: 7 days—>5 years), non-biodegradable (e.g., dieldrin soil persistence = 100 days), endocrine-disrupting (e.g., triclosan), and recalcitrant (e.g., dichlorodiphenyltrichloroethane) [1].

In contrast, certain ECs are identified as biomarkers as they are found in several water resources, while others can be chosen as model molecules due to their representative characteristics regarding a specific group of ECs [3]. Caffeine and triclosan have been considered within this group. In wastewater, caffeine and triclosan have been found in concentrations ranging 20–300 and 0.5–300,000 µg/L, respectively. Moreover, caffeine and triclosan were identified in the wastewater effluents from treatment plants (0.1–20.0 and 0.1–0.6 µg/L) and in surface water (10–80 and 50–2300 ng/L) [1]. It is presumed that their concentrations, along with those of other ECs, could have increased following the SARS-CoV-2 pandemic [4].

Caffeine is widely used in beverages (such as coffee, tea, and energy drinks), food, and certain medications [5]. On the other hand, triclosan is included in personal care products like detergents, cosmetics, and soaps due to its disinfecting properties [6]. Caffeine and triclosan differ in their physicochemical properties. Caffeine has high solubility in water (21,600 mg/L at 25 °C) and low bioaccumulation/toxicity (log K_ow_ = −0.07). Conversely, triclosan has low solubility in water (10 mg/L at 20 °C), high hydrophobicity, and a propensity for bioaccumulation (log K_ow_ = 3.5–4.8) [7,8]. Although caffeine is generally considered to have low toxicity, it has been detected in fish, frogs, and specific fruits and vegetables, as indicated by toxicological studies. Furthermore, similar studies have demonstrated its impact on reducing the growth rate in zebrafish embryos [7]. In contrast, triclosan, known for its toxic nature, impedes plant growth and soil respiration [1]. Chronic exposure to triclosan also triggers various toxic effects, including oxidative stress, neurotoxicity, genotoxicity, behavioral alterations, and histological deformities [6]. These attributes underscore the importance of removing caffeine and triclosan from wastewater.

Conventional treatment methods and natural attenuation are incapable of effectively removing all types of ECs from wastewater. In contrast, adsorption using agricultural residues has emerged as a promising and cost-effective alternative for wastewater treatment compared to other technologies/methods like membrane bioreactors (0.11–0.20 USD/m^3^ in China) [9]. Agricultural residues have demonstrated high efficiency in removing ECs (up to 100%) [3] and other contaminants, such as heavy metals (>90%) and organic matter (>80%) [1,10,11]. Several agricultural residues, including bagasse, straw, vegetable peels, fruit peels (e.g., banana, orange, coconut, coffee), seeds (e.g., peach stone, moringa), rice husk, and peanut shells, have been explored as adsorbents for ECs [10,11].

Agricultural residues are abundant and often underutilized, as exemplified by the case of peanut shells, which amount to a total of 11 million tons/year worldwide [12]. Given the large production of peanut shells, coupled with their irregular morphology boasting an approximate surface area of 2.01 m^2^/g and significant levels of lignin (27.0–33.0%), cellulose (34.7–45.0%), and hemicellulose (approximately 9.0%), peanut shells and other lignocellulosic residues demonstrate substantial potential as alternative adsorbents [13,14]. While peanut shells have primarily been employed for removing metals (e.g., adsorption capacity for cadmium = 6 mg/g) and dyes (e.g., adsorption capacity for reactive black 5 = 50 mg/g), they have also shown the ability to adsorb antibiotics like sulfathiazole and sulfamerazine, achieving adsorption capacities of 18.2 µg/g and 11.7 µg/g, respectively [13,15,16].

Despite the demonstrated efficiency of batch adsorption processes in removing ECs, their main limitation is associated with the ability to treat lower water volumes. Therefore, for adsorption to become a competitive process, the use of continuous treatment systems, such as fixed-bed columns, is necessary [17]. In these systems, agricultural residues have also proven to be effective in removing several contaminants. Residual yam pulp (Cr = 28.0 mg/g, Ni = 28.0 mg/g), residual banana pulp (Cr = 18.3 mg/g, Ni = 22.1 mg/g) [18], activated carbon from chichá-do-cerrado fruit husks (caffeine = 83.9 mg/g) [17], coconut shell-activated carbon (caffeine = 5.3 mg/g, saccharin = 4.3 mg/g, sulfamethoxazole = 6.2 mg/g, sucralose = 2.5 mg/g) [19], and peanut shell-activated carbon (naphthenic acids = 884 mg/g) [20] have been used. The efficiency of fixed-bed columns is related to the operational conditions and hydraulic performance of the fixed-bed column (flow rate, bed height, particle size of the bed material), making it highly desirable to optimize these parameters [17].

Nevertheless, given the limited use of raw peanut shells in removing ECs with varying physicochemical characteristics and favorable adsorption properties, this study aims to evaluate their adsorption capacity in removing caffeine and triclosan from synthetic solutions through batch and continuous adsorption tests.

## 2. Results and Discussion

### 2.1. Peanut Shell Characterization

#### 2.1.1. Physicochemical Composition

Table 1 displays the composition of peanut shells, including results from previous authors. The comparison of the physicochemical characteristics of peanut shells determined in this work with those from previous studies reveals variations in cellulose content (1.7–2.8 times lower), while the lignin and hemicellulose contents are higher, with increases of up to 1.3 and 2.9 times, respectively [13,14,21]. These variations can be ascribed to different factors, including the growth conditions of peanuts, diverse climatic influences, the level of product maturity at the time of consumption, processing conditions, among others [10].

In the case of the shells used in this study, it is observed that lignin stands out with the highest concentration, surpassing cellulose and hemicellulose by 16.8% and 10.1%, respectively [24]. Lignin, as cellulose, is notable for its abundance of functional groups, including hydroxyls, methoxyls, carbonyls, carboxyls, and sulfonates. These functional groups facilitate interactions with contaminants, thereby enhancing their adsorption [2]. Furthermore, collectively, cellulose, hemicellulose, and lignin, constitute the cell wall of peanut shells, forming a porous structure that would allow interactions between caffeine and triclosan solutions with different elements of the cell wall [25].

On the other hand, the extractives found in peanut shells have a lower concentration (between 1.3–2.4 times), compared to polysaccharides. This low concentration can be beneficial because the presence of extractives could reduce pore availability and potentially interfere with the adsorption of caffeine/triclosan. In fact, a higher content of extractives is associated with a decrease in contaminant adsorption [26].

The chemical composition analysis of peanut shells was supplemented by thermogravimetry analysis (TGA), which revealed three distinct weight loss regions. The initial phase, accounting for 10.8% of the overall weight loss, occurs between 20 and 122.6 °C and is linked to the evaporation of moisture, particularly the water content. Subsequently, a significant weight loss (making up 52.1% of the total) takes place in the temperature range of 122.6 to 409.5 °C, possibly indicating the decomposition of hemicellulose, cellulose and a portion of lignin components. The final stage of weight loss (410.3–600 °C) results in a cumulative weight reduction of 72.1% and can be attributed to the total degradation of lignin and the carbonaceous material [27].

#### 2.1.2. Point of Zero Charge

The pH significantly influences the adsorption process by affecting the charge of the adsorbent and the ionization of the adsorbate. Peanut shell’s pH_pzc_ remains consistent across different particle sizes, measuring 7.5, 7.6, and 7.4 for small, medium, and large particles, respectively, as depicted in Figure 1. Below the pH_pzc_, the positively charged peanut shells effectively adsorb negatively charged contaminants, while at higher pH levels, their negative charge enables the adsorption of positively charged contaminants. Caffeine and triclosan exhibit pKa values of 10.4 and 7.9–8.1, respectively, suggesting enhanced removal efficiency at pH levels below the pH_pzc_ [7,28,29]. At acidic pH, both contaminants become neutral, mitigating repulsive electrostatic forces and thereby improving adsorption [8]. Consequently, operating at a pH of 6.5 (±0.2) would optimize the removal of caffeine and triclosan.

#### 2.1.3. FTIR Analysis

Figure 2 shows the FTIR spectra of peanut shells, both before and after the adsorption of contaminants. In the spectra of peanut shells prior to adsorption, the initial band at 3350 cm^−1^ represents the OH− groups present in lignin, cellulose, and hemicellulose. At 2920 cm^−1^, a band related to the C−O stretching appears, while bands at approximately 1700 and 1369 cm^−1^ are attributed to carboxyl and hydroxyl groups, respectively. The C−O bonds from cellulose are evident at 1259 cm^−1^. The presence of lignin, characterized by the vibration of its aromatic rings, is associated with the bands at 1620, 1509, and 1422 cm^−1^. The band at 1024 cm^−1^ is linked to the C−H_2_ group in cellulose, and the band at 1369 cm^−1^ corresponds to C−H bonds present in both cellulose and hemicellulose [13,30].

The FTIR spectra of peanut shells for three different particle sizes after the adsorption of caffeine exhibit minimal differences among them. However, there is a noticeable decrease in transmittance from 1530 to 1183 cm^−1^ when compared to the FTIR spectra of peanut shells before caffeine adsorption. This reduction is particularly pronounced in the bands at 1369 cm^−1^ (associated with cellulose), 1509 cm^−1^ (related to hemicellulose), and 1620, 1509, and 1422 cm^−1^ (attributed to lignin). These reductions in transmittance suggest an affinity between these functional groups and caffeine. Notably, the 1620 cm^−1^ band appears slightly broader, possibly due to the presence of a caffeine band at 1642 cm^−1^. A similar broadening effect is observed in the 1024 cm^−1^ band [13,30]. Nevertheless, caffeine bands from 1024 to 1548 cm^−1^ may be more intense in the small particles, likely because the amount of caffeine adsorbed by the peanut shells is higher for this particle size.

Similarly, the FTIR spectra of peanut shells with different particle sizes following triclosan adsorption exhibit striking resemblances. The primary distinction between the spectra before and after adsorption lies in a subtle reduction in transmittance. Changes in the bands at 3350 and 2920 cm^−1^ are evident, likely indicating their interaction with the triclosan molecule. This phenomenon is also observed in the bands spanning from 1024 to 1700 cm^−1^, suggesting the occurrence of an adsorption process. Furthermore, these bands correspond to the functional groups associated with cellulose, hemicellulose, and lignin, all of which are integral to the adsorption process [2].

#### 2.1.4. Morphological Characterization

Figure 3 shows the SEM images of small, medium, and large particles. They reveal an irregular surface morphology and the presence of pores, consistent with the findings of Li et al. [13]. These morphological features indicate favorable adsorption properties, which are conducive to the removal of caffeine and triclosan.

The specific surface area of the peanut shells, as determined by a BET multipoint test, was found to be 1.7 m²/g. This value is lower than those reported in previous studies (2.4 and 5.0 m²/g) [21,22]. The variance is likely due to differences in peanut species and particle size (177–250 µm and 250–500 µm) [18,23] The pore volume and pore diameter, determined using the Barrett–Joyner–Halenda (BJH) method, were 0.005 cm³/g and 3.14 nm, respectively. According to the IUPAC, a pore diameter within the range of 2 to 50 nm characterizes mesoporous materials [31]. The nitrogen adsorption–desorption isotherm for the studied peanut shells, as shown in Figure 3, corresponds to a type III isotherm according to the Brunauer–Deming–Deming–Teller classification. This type indicates a macroporous or non-porous material with limited affinity between the adsorbate and the adsorbent. Once a molecule is adsorbed, it behaves as if it is free to adsorb another molecule, with certain parts acting as monolayers and others as multilayers [32].

### 2.2. Adsorption Tests

Figure 4a,b illustrate the removal of caffeine and triclosan using several doses of small, medium, and large peanut shell particles, respectively. Optimal doses of peanut shells for caffeine removal (Figure 4a) were determined to be 10 g/L for small particles, 17 g/L for medium particles, and 35 g/L for large particles. These doses achieved removal efficiencies of 85.6 ± 1.4%, 84.1 ± 0.1%, and 87.8 ± 2.7% and adsorption capacities of 3.7 ± 0.1 mg/g, 31.9 ± 0.3 mg/g, and 1.0 ± 0.3 mg/g, respectively. In the case of triclosan removal (Figure 4b), the optimal doses for small, medium, and large particles were 0.1 g/L, 1.0 g/L, and 10.0 g/L, leading to removal efficiencies of 89.3 ± 1.5%, 90.8 ± 0.3%, and 90.9 ± 4.6% and adsorption capacities of 238.9 ± 2.0 mg/g, 31.9 ± 2.2 mg/g, and 3.9 ± 0.2 mg/g, respectively.

A higher dosage of peanut shells is required to remove caffeine/triclosan as the particle size of the peanut shell increases. This is primarily because smaller particles offer a greater surface area available for caffeine/triclosan adsorption [14]. As a result, smaller particles prove to be the most effective for the removal of both contaminants. However, when higher quantities of peanut shells were used, the aqueous solution’s color darkens due to substances released by the peanut shells [26]. This issue can lead to competitive adsorption between water, the compounds released from the peanut shells, and caffeine/triclosan [9].

Figure 4c,d depict removal of caffeine and triclosan for the three different particle sizes of peanut shells, employing the optimal doses at different contact times. The optimal contact time for caffeine removal with all three particle sizes of peanut shells was found to be 180 min, resulting in removal of 85.0 ± 1.4%, 89.7 ± 0.3%, and 88.8 ± 1.2% for small, medium, and large particles, respectively. In contrast, the optimal contact time for triclosan adsorption with all three particle sizes of peanut shells was determined to be 45 min, leading to removal of 90.1 ± 2.1%, 89.2 ± 2.7%, and 94.9 ± 1.4% for small, medium, and large particles, respectively. During the initial stages of the adsorption process, the highest rates of caffeine (30 min) and triclosan (15 min) adsorption occur. This is because the active sites were initially unoccupied, leading to rapid adsorption on the external surface of the peanut shells. Subsequently, a slower adsorption process took place within the pores until equilibrium is achieved [33].

Furthermore, a higher affinity between triclosan and peanut shells was observed due to triclosan’s greater hydrophobicity (log K_ow_ = 4.76). Triclosan’s high hydrophobicity results in limited solubility in water (10 mg/L at 20 °C) and the potential for bioaccumulation in peanut shells [23]. This explains why the doses of peanut shells and contact times required to remove triclosan are lower than those needed for the removal of the same concentration of caffeine.

The statistical analysis indicates significant differences in the removal efficiencies and adsorption capacities of both contaminants, which are influenced by the size of peanut shell particles and the type of contaminant (*p*-value > 0.05). This finding underscores the importance of these factors in the adsorption process, potentially offering valuable insights for optimizing remediation strategies in practical applications.

### 2.3. Adsorption Kinetics and Isotherm Models

Table 2 and Figure 5a,b show data for the non-linear pseudo-first- and pseudo-second-order kinetics models for caffeine and triclosan adsorption. According to the values of R^2^, χ^2^ and SSE for the three particle sizes, the model that best fits the removal of both contaminants is the pseudo-second-order model. The maximum adsorption capacity in the adsorption kinetics tests for caffeine and triclosan was obtained with the small particles, reaching values of 3.2 and 289.3 mg/g, respectively. The fitting to the pseudo-second-order kinetics model complemented with the several functional groups presented in peanut shells (Figure 1) suggest a chemical adsorption. This indicates that speed control is given by reactions between peanut shells and caffeine/triclosan, rather than by a mass transfer. This involves the exchange of electrons between the OH− or ligand groups and the contaminants [25].

As in previous studies, in which peanut shells were used to remove sulfonamides [13], PB5 dye [15] and some metals such as Cu (II), Cd^2+^, Hg^2+^, Pb (II) [14,30], the kinetics models were also fitted to pseudo-second-order kinetics. The same fitting was obtained using adsorbents such as thermally modified verde-lodo bentonite [34], oxidized biochar from pine needles [5], rice husk, coconut fiber, corn cob [10] in the removal of caffeine, and kenaf-derived biochar [8], activated carbon from coconut pulp waste [35] and char from palm kernel shells [6] in the removal of triclosan. The authors conclude that the adsorption of both contaminants occurs by chemisorption [5,6,11,35].

Nonetheless, according to other authors, merely applying the pseudo-second-order model does not offer a complete grasp of the adsorption mechanisms. Indeed, some studies propose physisorption as the primary process in caffeine and triclosan removal, predominantly characterized by π–π stacking, hydrogen bonding, and electrostatic attractions [8,36].

The parameters of the intraparticle diffusion model were also detailed in Table 2. This model provides a valuable tool for identifying reaction pathways, adsorption mechanisms, and anticipating the step that controls the adsorption rate. In the solid-liquid adsorption process, the transfer of the contaminant is commonly described by external diffusion, also known as film diffusion, surface diffusion, and pore diffusion, or a combination of these phenomena [2]. This is graphically illustrated by analyzing the Qt vs. t^0.5^ curve; if this curve passes through the origin, it indicates that the adsorption process is solely restricted by intraparticle diffusion. If not, the mechanism may be controlled by multiple processes [37]. In this study (Figure 6), the presence of two linear regions is observed, suggesting that the adsorption process is governed by a two-step mechanism. In the first stage, contaminants (caffeine/triclosan) migrate from the liquid phase to the external surface of peanut shells through the hydrodynamic boundary layer (film diffusion). In the second stage, there is slower diffusion (intraparticle diffusion) of caffeine/triclosan molecules from the external surface of peanut shells to their pores, where they are adsorbed.

Table 3 and Figure 5c,d show the data calculated for the Langmuir, Freundlich, and Sips non-linear isothermal fittings. Triclosan adsorption exhibited the best fit with the Sips model across small (120–150 µm), medium (300–600 µm), and large (800–2000 µm) particles, with corresponding R^2^ values of 0.997, 0.973, and 0.989, and maximum adsorption capacities (q_m_) of 238.98, 29.45, and 7.38 mg/g, respectively. This is corroborated with the SSE and χ^2^ values, which are the lowest values among the three isothermal models applied. The Sips model combines the Freundlich and Langmuir models, considering the heterogeneity of peanut shells as in the Freundlich model and the monolayer formation of the Langmuir model. In addition, at the beginning of the adsorption, the model will resemble that of Freundlich, with no apparent limit on the adsorption capacity of triclosan. As the adsorption continues, the model with the best fit is that of Langmuir. It was observed that the experimental triclosan adsorption data also fit the Sips model when using activated carbon from coconut shells (raw and modified) [38], gas masks [39], and empty palm cluster biochar [40].

For the caffeine adsorption, R^2^ values close to 1 (0.84–0.995) were obtained for the three isotherm models used. Small and large particles have a better fit to the Freundlich model while medium particles fit to the Langmuir and Sips models. However, considering the value of 1/n calculated with the Freundlich model (less than 1) and the fitting of the experimental data (experimental q_e_) when using more concentrated caffeine solutions (closer to the Langmuir model), it was determined that the correct fit is that to the Sips model. Values of 1/n greater than 1 indicate a cooperative adsorption and those smaller than 1 correspond to a monolayer adsorption [41]. These results would indicate a combination of the Freundlich (low concentrations of caffeine) and Langmuir (high concentrations of caffeine) models. Lower values of SSE and χ^2^ for the Sips model confirm that this is possible. Similar results were obtained when using grape stalk, grape stalk chemically modified, and activated carbon from grape stalk in the caffeine removal [42].

Most of the research previously performed using peanut shells was carried out to remove heavy metals and dyes. As mentioned before, there are only a few studies performed to remove ECs [13]. In those studies, removals between 32.8 and 72.8% and adsorption capacities between 8.2 and 18.2 µg/g were achieved in the removal of sulfamerazine, sulfamethazine, sulfathiazole and sulfamethoxazole using 500–600 µm peanut shell particles. The adsorption of the four contaminants was fitted to the Freundlich and the pseudo-second-order model. Meanwhile, Tomul et al., 2020, used particles with a 150–250 µm size and achieved adsorption capacities of 56.1 and 24.5 mg/g for the removal of diclofenac and bisphenol, respectively [43].

Even though caffeine/triclosan removal using peanut shells has not been sufficiently studied, it is possible to mention that the adsorption capacities of peanut shells obtained are comparable or even higher than those reached with other agricultural residues. Triclosan removal using activated carbon from coconut pulp (595 µm) was fitted to a pseudo-second-order kinetics model and to the Langmuir isotherm (q_m_ = 38.91 mg/g) [44]. These results are similar to those obtained with the medium size peanut shell particles even though raw (untreated) peanut shells are used. Likewise, in the study carried out by Triwiswara et al. [45], the triclosan removal using rice husk treated at 300 °C (150–500 µm) was fitted to the pseudo-first-order kinetics and to Langmuir model (q_m_ = 72.7 mg/g). This adsorption capacity is lower compared to that obtained with small peanut shell particles. In the case of caffeine, the adsorption capacity obtained using hydrochars from pistachio shells (500–1000 µm) and the Langmuir model was slightly higher (5.5 mg/g) than the capacity obtained with large particles of peanut shells [46]. Meanwhile, using grape stalk with a mean particle size of 700 µm, the adsorption data best fitted the Sips model with a q_m_ = 89.2 mg/g, which was much higher than the value obtained in this work [42].

### 2.4. Fixed-Bed Columns

The results from the experiments assessing the efficiency of removing caffeine and triclosan in fixed-bed columns with various particle sizes are outlined in Table 4.

There were no statistically significant differences (*p*-value > 0.05) in the removal efficiencies of both contaminants across different bed heights (4, 5, 8 cm). However, column clogging occurred with the smallest particles under the highest hydraulic load (4 m^3^/m^2^-day) after just 20 min of operation, resulting in solution spillage. Consequently, it was not possible to construct the breakthrough curve. Hence, Figure 7 displays the experimental breakthrough curves for the three particle sizes of peanut shell, using a 4 cm bed height and a hydraulic load rate of 2 m^3^/m^2^-day. The breakthrough curves for all three particle sizes and both contaminants exhibit a consistent S-shaped pattern, suggesting that the adsorption process is primarily influenced by mass diffusion resistance and surface bonding within the peanut shell particles [47].

The significant influence of peanut shell particle size on column hydraulic behavior is readily evident. This parameter governs inter-particle spacing within the adsorbent material, thereby directing the path traversed by contaminant-laden solutions [48]. Moreover, particle size significantly influences the efficiency of contaminant removal over time. Specifically, observations indicate that larger particles (800–2000 µm) demonstrate diminished adsorption capacities for caffeine (q_b_ = 2.39–3.14 and q_s_ = 1.61–1.74 times) and triclosan (q_b_ = 2.25–4.45 and q_s_= 1.51–2.28 times) relative to smaller counterparts. This is attributed to the varying availability of active sites, as noted in the batch adsorption tests, which is directly linked to particle size. Smaller particles typically boast a higher number of active sites [49].

The influence of peanut shell particle size on filter bed exhaustion is an important parameter in filtration systems. Observations reveal that an increase in particle size correlates with heightened filter bed exhaustion, thereby impacting the treated water volume (V_b_) up to breakthrough. Larger particles correspond to lower treated water volumes, with significant reductions of up to 2.9-fold for caffeine and 3.7-fold for triclosan with the largest particles.

Furthermore, the effectiveness of the hydrodynamic mass transfer zone (h_MTZ_) is closely tied to particle size. Larger particles demonstrate diminished mass transfer efficiency, resulting in a wider h_MTZ_ [50]. Notably, for caffeine and triclosan, the h_MTZ_ expands by up to 2.8-fold and 2.7-fold, respectively, with larger particles. This is attributed to the fact that larger particles, with reduced adsorption capacity, necessitate a larger volume for achieving efficient mass transfer [49]. Verification of the largest mass transfer zone is achieved by analyzing the slope of the C_f_/C_0_ vs. t graph (refer to Figure 5), where a steeper slope indicates larger particles [50].

Additionally, the augmented h_MTZ_ in larger particles precipitates a decline in fractional bed utilization (FBU), translating to a reduction between 12.8% and 24.1% for caffeine, and between 13.4% and 40.0% for triclosan. The increased interstitial spaces among larger particles elevate the void fraction within the bed, thereby diminishing the effective contact time between caffeine/triclosan and peanut shell particles [48].

Furthermore, apart from particle size, the chemical nature of contaminants plays an important role in material efficiency. Peanut shells demonstrate superior adsorption capacity for triclosan relative to caffeine. This discrepancy stems from the physicochemical properties of the contaminants. While caffeine exhibits high hydrophilicity, resulting in a pronounced affinity for the aqueous phase, triclosan’s lipophilic nature fosters a stronger interaction with peanut shells [1].

Table 5 shows the results of the analysis of experimental data on adsorption in fixed-bed columns, which were fitted to the Bohart–Adams, Thomas, and Yoon–Nelson kinetics models. The three models are mathematically equivalent, reflected in consistent values of R^2^ (caffeine: 0.976–0.980, triclosan: 0.949–0.970), SSE (caffeine: 0.055–0.061, triclosan: 0.076–0.217), and χ^2^ (caffeine: 0.007–0.011, triclosan: 0.009–0.022) for each particle size. This means they provide the same symmetrical breakthrough curve [51]. In the Bohart–Adams model, the rate of binding is directly related to the residual binding capacity of the peanut shell adsorbent bed and the concentration of caffeine/triclosan. This model focuses on describing adsorption in continuous flow systems, particularly in the initial phase of the decomposition curve, where the ratio of contaminant concentration in the effluent to the initial concentration is less than 0.5. During this stage, insignificant mass transfer of the contaminant from the aqueous medium to the adsorbent is assumed [52]. The trend shows that K_AB_ increases with larger particles, up to 3.3 times for caffeine and 2.9 times for triclosan, while with N_0_, its value decreases with the increasing particle size of the peanut shell, up to 3.5 times for caffeine and 2.7 times for triclosan. Particle size significantly impacts contact time, which is dictated by the material’s morphology and the spacing between particles. Larger peanut shell particles, owing to their greater individual volume, produce wider gaps between them, thereby accelerating flow through the column and reducing contact time. Nevertheless, larger particles also entail a reduced active surface area, consequently diminishing the material’s adsorption capacity. Conversely, smaller particles yield the opposite effect [19].

On the other hand, the Thomas model is based on a Langmuir-type adsorption process with reversible second-order kinetics. In this model, the adsorption rate is not controlled by internal or external mass transfer but by mass transfer at the interface, making it useful for evaluating the relationship between concentration and time in breakthrough curves [19,52]. The results indicate a higher adsorption capacity q_0_ for the smaller adsorbent in both caffeine and triclosan removal, up to 3.5- and 120.8-fold, respectively. However, the value of the Thomas constant k_TH_ decreases as the particle size decreases. This implies an increased effect of mass transfer in the adsorption process, and the adsorption of caffeine/triclosan becomes kinetically favorable as the particle size of peanut shells decreases [47]. Additionally, there is a good fit to the model, indicating that external and internal diffusion are not limiting steps in the process [52].

Finally, the Yoon–Nelson model is used to determine the probability of adsorption for each caffeine/triclosan molecule, which is proportional to their adsorption capacity and probability of advancement in the adsorbent. This model considers both intraparticle and external mass transfer, as well as resistance to diffusion in the boundary layer and solid–liquid interface, proving useful for understanding adsorption kinetics in fixed-bed systems and optimizing adsorption column design [52]. The results of fitting to this model indicate a directly proportional relationship between k_YN_ and particle size, while an inverse relationship is observed between particle size and τ for both contaminants. The results of fitting to this model for both contaminants indicate that the time required to retain 50% of the initial caffeine and triclosan was higher for the smallest peanut shell particles (153.7 and 667.2 min, respectively) and lower for the larger ones, while k_YN_ has the lowest values for the smaller particles and the highest for the larger ones. This suggests a directly proportional relationship between k_YN_ and particle size, while an inverse relationship is observed between particle size and τ. The last is due to the higher surface area that show the lowest particle [49].

Despite the good fit of the experimental data for all three particle sizes, differences between caffeine and triclosan were observed across all models, indicating a higher adsorption capacity for the latter, which is consistent with the batch tests. Yanala and Pagillab (2020) attribute this behavior to differences in their physical-chemical characteristics [53].

### 2.5. Alternatives for Saturated Peanut Shells

The regeneration of saturated adsorbents is a pivotal alternative, particularly for adsorbents that are not readily available or whose production is complex and costly, such as activated carbon [54]. Common regeneration methods face limitations at an industrial scale due to high energy consumption or the use of chemicals such as acids, bases, chelators, and/or supercritical fluids [55]. Additionally, regenerated adsorbents often lose efficiency with each regeneration cycle, typically maintaining up to 80% of their original capacity. However, their lifespan can be extended if the saturated adsorbent is used for the removal of other contaminants, utilizing the functional groups left by the initially removed contaminant [56,57]. Nevertheless, the saturated adsorbent eventually becomes waste, posing the challenge of finding alternatives for its subsequent use or final disposal to prevent secondary environmental pollution and reduce wastewater treatment costs [58].

One promising alternative is converting the saturated adsorbent into biochar, as the resulting material can be used as a new adsorbent or as a soil conditioner [55,58]. Moreover, if the saturated adsorbents were used in the removal of heavy metals, they can be repurposed as supercapacitors, catalysts, or catalyst supports, thereby replacing costly materials such as carbon nanotubes [5,6]. Additionally, saturated adsorbents can be efficiently utilized as reinforcements in composite materials, improving the properties of the composites [59].

Reusing saturated adsorbents is of paramount importance to enhance sustainability and reduce waste. It is worth noting that in our study, we used peanut shells, which are highly available. Therefore, we did not pursue the regeneration of the peanut shells used as adsorbents. This decision was based on their abundance and the practicality of utilizing them without the need for complex conditioning and regeneration processes.

## 3. Materials and Methods

### 3.1. Materials

Peanut shells were obtained from a peeling plant located in the city of Loja (southern Ecuador, −3.99313 and −79.20422). Caffeine (CAS: 58-08-2) and triclosan (CAS: 3380-34-5) used for the synthetic solution preparation were ReagentPlus^®^, Sigma-Aldrich brand (St. Louis, MI, USA), 99.0% and ≥ 97.0% pure, respectively.

#### Peanut Shell Conditioning

Peanut shells underwent a thorough cleaning process involving washing with both tap and distilled water to eliminate impurities. Following this, the pristine shells were dried at 80 °C for 24 h and subsequently crushed using a Thomas knife mill, model 3379-K05 (Thomas Scientific, Swedesboro, NJ, USA) [14]. The resulting material was then carefully sieved and sorted into three distinct size categories: small particles (125–150 µm), medium particles (300–600 µm) and large particles (850–2000 µm).

### 3.2. Experimental Model

#### 3.2.1. Batch Adsorption Tests

The batch adsorption tests were conducted under dark conditions to prevent degradation processes, and each test was performed in triplicate using a control sample. Distilled water served as the control sample for caffeine, while a 5% *v*/*v* NaOH solution was used as the control sample for triclosan. The control sample was utilized to mitigate errors arising from the leaching of water-soluble components of peanut shells, particularly when higher doses were used.

The optimal dosage of peanut shells was determined by employing 20 mL of caffeine/triclosan solutions with a concentration of 30 mg/L. The solutions were stirred at 150 rpm for 180 min and 45 min, respectively. The doses of peanut shells for the removal of caffeine and triclosan were set as follows: 5–60 and 0.1–35 g/L for large particles, 3–40 and 0.1–1.5 g/L for medium particles, and 1–20 and 0.02–1.0 g/L for small particles. Subsequently, the ideal contact time and adsorption kinetics was established using the optimal dose for each particle size. The peanut shells were immersed in caffeine/triclosan solutions for periods ranging from 0 to 300 min and 0 to 120 min, respectively.

Adsorption isotherms were derived under ideal conditions, including optimal dosages and contact durations, using caffeine and triclosan solutions spanning seven concentrations (5–50 mg/L). Each batch adsorption experiment was replicated three times across three particle size ranges, while maintaining a consistent pH of 6.5 (±0.2) and temperature (room temperature: 22.7 (±1.1) °C). The efficacy of caffeine/triclosan removal served as the focal variable under examination in these batch adsorption trials. Finally, separation of the peanut shells from the aqueous solution was accomplished using cellulose filters with a pore size of 0.2 µm [13].

#### 3.2.2. Fixed-Bed Column Tests

Fixed-bed column tests were conducted for all three particle sizes. Columns measuring 10 cm in height and 1 cm in diameter were employed, and they were filled with 4, 5 and 8 cm of peanut shells. Prior to the operation of the columns, the shells were washed with distilled water until the color in the washing water was completely eliminated. The columns were fed with caffeine/triclosan solutions of 30 mg/L, applying hydraulic loading rates of 2 m³/m²-day (0.016 m³/h) and 4 m³/m²-day (0.032 m³/h). These flow rates are consistent with those used with biochar at lab scale for the removal of some emerging contaminants [60].

### 3.3. Analytical/Instrumental Methods

#### 3.3.1. Peanut Shell Characterization

Moisture [61] extractives [62,63], lignin, cellulose and hemicellulose [64,65], ash [66] and volatile material [67] of peanut shells were determined according to ASTM standards. In addition to the physicochemical characterization, thermogravimetric analysis was conducted using a Shimadzu thermo-balance, model 50, over a temperature range from 20 to 600 °C. The analysis featured a heating rate of 10 °C/min and was performed under a nitrogen atmosphere (at a flow rate of 50 mL/min).

The point of zero charge, or pHpzc, for peanut shells was determined by employing 50 mL of distilled water. The pH was adjusted within the range of 2 to 11 using 0.01 M NaOH and HCl solutions. Following this, 0.5 g of peanut shells were added to each solution, and the samples were then shaken at 150 rpm for 48 h. This experiment was replicated for each of the three particle sizes, with duplicate tests conducted for accuracy [28].

To assess the functional groups, present in caffeine, triclosan, and the peanut shells (both before and after the adsorption tests), a FTIR-6800 spectrometer (Perkin Elmer, MA, USA) equipped with a diamond crystal ATR was employed. Furthermore, the morphology of the peanut shells across the three distinct particle size categories was examined using an ASPEX PSEM eXpress scanning electron microscope (SEM, ASPEX Corporation, Delmont, PA, USA). The SEM operated with a 20.4 mm working distance and a 15 keV acceleration voltage. For the determination of specific surface area and pore radius of the peanut shells, physical nitrogen adsorption was employed at a bath temperature of 77 K, and the Brunauer–Emmett–Teller (BET) method was used. The pore volume was calculated using the Barret–Joyner–Halenda (BJH) model. To prepare the peanut shells for this analysis, they were heated to 100 °C at a rate of 10 °C/min and held under vacuum for 1440 min.

#### 3.3.2. Caffeine and Triclosan Concentrations before/after Adsorption Tests

To determine the concentrations of caffeine and triclosan, an Analytik Jena Spercord 210 Plus UV-vis spectrophotometer (Analytik Jena AG, Jena, Germany) was utilized, performing a scan between 190 and 800 nm to identify wavelengths with the highest absorbance. A solution of 20 mg/L was employed for both caffeine and triclosan during this procedure.

Following this, calibration curves were established using solutions within a concentration range of 1 to 50 mg/L for caffeine (dissolved in distilled water) and triclosan (dissolved in a 5% *v*/*v* NaOH solution). The resulting adjustment equations were as follows: for caffeine, C = 0.0153A + 0.0185, with a coefficient of determination (R^2^) of 0.994; and for triclosan, C = 0.0064A − 0.0022, with an R^2^ of 0.996. In these equations, C represents the concentration of caffeine/triclosan, and A represents absorbance.

### 3.4. Data Analysis

#### 3.4.1. Adsorption Kinetics and Isotherm Study

Adsorption kinetics describes the adsorption rate of caffeine/triclosan and provides information about reaction pathways [25]. Adsorption kinetics of caffeine and triclosan were calculated using data obtained to determine the optimal contact time. The amount of the caffeine/triclosan adsorbed per gram of peanut shells was calculated using Equation (1).
(1)$$q=\frac{\left(C_{o}-C_{f}\right)V}{W}$$
where q (mg/g) is the concentration of caffeine/triclosan per gram of peanut shells, W (g) is the mass of peanut shells used, and V (L) is the caffeine/triclosan solution volume. The obtained data were fitted to non-linear pseudo-first-order, pseudo-second-order and Elovich models. Equations (2)–(4) were used, respectively [25].
(2)$$q_{t}=q_{e}(1-e^{-K_{1}t})$$
(3)$$\frac{t}{q_{t}}=\frac{1}{K_{2}q_{e}^{2}}+\frac{t}{q_{e}}$$
(4)$$q_{t}=\frac{1}{\beta }ln(1+\alpha \beta t)$$

Moreover, the intraparticle diffusion model (Equation (5)) was applied to obtain information about the mechanism acting in the caffeine/triclosan adsorption process.
(5)$$q_{t}=K_{p}\sqrt{t}+C$$
where K_1_ (min^−1^) is the pseudo-first-order rate constant, K_2_ (g/mg min) is the pseudo-second-order rate constant, q_e_ (mg/g) represents the caffeine/triclosan adsorbed on equilibrium and q_t_ (mg/g) is the amount of caffeine/triclosan adsorbed at time t, α is the initial rate constant, (mg/g min), β (mg/g) is the desorption constant, K_p_ (mg/g min^1/2^) is the rate constant of the intra-particle diffusion model, and C (mg/g) is a constant associated with the thickness of the boundary layer.

The development of an isotherm model is essential for designing and optimizing adsorption processes. The adsorption isotherms were obtained with the optimal adsorption conditions corresponding to each particle size. Caffeine/triclosan solutions of 10, 20, 30, 40 and 50 mg/L were used. The obtained data were fitted to non-linear Langmuir, Freundlich and Sips models. The equation models are presented in Equations (6)–(8), respectively [25]:(6)$$q_{e}=\frac{q_{m}k_{L}C_{e}}{1+k_{L}C_{e}}$$
(7)$$q_{e}=k_{F}C_{e}^{1/n}$$
(8)$$q_{e}=\frac{q_{m}{\left(k_{S}C_{e}\right)}^{1/n}}{1+{\left(k_{S}C_{e}\right)}^{1/n}}$$
where q_e_ (mg/g) is the amount of caffeine/triclosan per unit mass of peanut shells in equilibrium, C_e_ (mg/L) is the liquid phase concentration of caffeine/triclosan in equilibrium, q_m_ (mg/g) is the maximum adsorption capacity of peanut shells, k_L_ (L/mg) is the Langmuir constant related to adsorption energy, k_F_ ((mg/g)/(mg/L)^n^) is the Freundlich capacity constant, 1/n is the Freundlich intensity parameter, and k_S_ (L/mg) is the Sips constant related to micropores energy [37].

The kinetics and isotherm models used were selected because they have typically presented good fit (R^2^ close to 1), when agricultural residues are used in the removal of several contaminants (dyes, heavy metals, ECs) [2,37].

#### 3.4.2. Breakthrough Curve Models

To assess the adsorption of contaminants, concentration curves of caffeine/triclosan in the effluent as a function of time were generated. Caffeine/triclosan adsorption at penetration time (q_b_) and saturation time (q_s_) (in mg/g) was calculated using Equations (9) and (10):(9)$$q_{s}=\frac{C_{o}*Q}{1000*m}\int_{0}^{t_{s}}\left(1-\frac{C_{s}}{C_{o}}\right)dt$$
(10)$$q_{b}=\frac{C_{o}*Q}{1000*m}\int_{0}^{t_{b}}\left(1-\frac{C_{b}}{C_{o}}\right)dt$$
where C_o_ (mg/L) is the initial concentration of caffeine/triclosan, C_b_ and C_s_ (mg/L) are the concentration of caffeine/triclosan at the breakthrough time and the saturation time, respectively, Q (mL/min) is the volumetric flow, m (mg) is the peanut shell mass, t_b_ (min) is the breakthrough time (C/C_o_ = 0.1) and t_s_ (min) is the saturation time (C/C_o_ = 0.9). Other parameters, as the empty bed contact time (EBCT), height of the mass transfer zone (h_MTZ_) (cm), and the percentage of fractional bed utilization (%FBU) were determined using Equations (11)–(13), respectively [52]:(11)$$EBTC=\frac{V_{c}}{Q}100$$
(12)$$\%FBU=\frac{q_{b}}{q_{s}}100$$
(13)$$h_{MTZ}=\left(1-\frac{q_{b}}{q_{s}}\right)h$$
where Q is the flow rate (L/d), Vc is the fixed-bed volume (L) and h is the bed height (cm).

The data obtained from the continuous tests were fitted to the non-linear Bohart–Adams, Thomas and Yoon–Nelson models, as expressed in Equation (14)–(16).
(14)$$\frac{C_{t}}{C_{o}}=\frac{1}{1+e^{\left(\frac{K_{BA}*N_{0}*h}{u}-K_{BA}*C_{0}*t\right)}}$$
(15)$$\frac{C_{t}}{C_{o}}=\frac{1}{1+e^{\left(\frac{k_{Th}}{Q}*(q_{0}m-C_{0}*V_{eff})\right)}}$$
(16)$$\frac{C_{t}}{C_{o}}=\frac{1}{1+e^{\left(k_{YN}*(\tau -t\right))}}$$
where C_t_ (mg/L) is the concentration at time t, t (min) is the service time of the column, K_BA_ (L/(min-mg)) is the rate constant of the Bohart–Adams model, N_o_ (mg/L) is the maximum adsorptive capacity, h (cm) is the bed height, and *u* (cm/min) is the linear flow velocity. k_Th_ is the Thomas rate constant (mg/min), V_eff_ is the volume of effluent (mL), q_0_ is the maximum caffeine/triclosan concentration (mg/g). k_YN_ is the Yoon–Nelson constant (min^−1^) and τ refers to the duration (min) required to retain 50% of the initial adsorbate.

### 3.5. Statistical Analysis

The optimal dosage, contact time and bed height were determined through an analysis of variance (ANOVA) with a single factor, analyzed using Tukey’s test (significance level of 95.0%). For the calculation of non-linear parameters, including isotherms (Langmuir, Freundlich, and Sips models), kinetics (pseudo-first- and -second-order and Elovich models), as well as the Bohart–Adams models, Origin Pro (version 2019) was employed. To determine the model that best fits the obtained data, Equations (17)–(19) were utilized to calculate squared errors (SSE), chi-squared (χ^2^), and the coefficient of determination (R^2^), respectively.
(17)$$R^{2}=1-\frac{\sum {\left(P_{e,exp}-P_{e,cal}\right)}^{2}}{\sum {\left(P_{e,exp}-P_{e,mean}\right)}^{2}}$$
(18)$$\chi^{2}=\sum \frac{{\left(P_{e,exp}-P_{e,cal}\right)}^{2}}{q_{e,cal}}$$
(19)$$SSE=\sum {\left(P_{e,exp}-P_{e,cal}\right)}^{2}$$
where P_e,exp_ are the experimental parameters: q (batch) and C_t_/C_o_ (fixed-bed columns), P_e,cal_ are the calculated parameter using solver tool, and P_e,mean_ is the mean of P_e,exp_ values (mg/g) [57].

## 4. Conclusions

The findings of this study underscore the promising potential of peanut shells as a cost-effective and environmentally sustainable adsorbent for removing caffeine and triclosan from wastewater. The characterization performed revealed that peanut shells are primarily composed of polysaccharides, with lignin, cellulose, and hemicellulose constituting approximately 70% of their composition. Experimental investigations demonstrated that smaller particles (120–150 µm) exhibited the highest removal efficiencies, achieving up to 85.6 ± 1.4% for caffeine and 89.3 ± 1.5% for triclosan removal. Notably, triclosan showed greater susceptibility to removal compared to caffeine, which was attributed to its lower water solubility and higher hydrophobicity. The empirical data further validated the pseudo-second-order kinetics model as the best fitting, suggesting a chemisorption mechanism governing the interaction between the contaminants and peanut shells. Moreover, the Sips model effectively captured the adsorption equilibrium, revealing maximum adsorption capacities of 3.3 mg/g for caffeine and 289.3 mg/g for triclosan. The results obtained from the fixed-bed column experiments highlighted the impact of particle size on adsorption efficiency and hydraulic behavior. Smaller particles exhibited the highest adsorption capacity; however, issues related to clogging were observed. Despite this, the experimental data exhibited good agreement with the Bohart–Adams, Thomas, and Yoon–Nelson models, further emphasizing the potential of peanut shells as an effective adsorbent in wastewater treatment processes. These findings underscore the importance of peanut shells as a sustainable and efficient alternative for mitigating the presence of emerging contaminants in wastewater treatment processes. Their ability to adsorb other hydrophilic compounds with characteristics similar to caffeine, such as salicylic acid and cephalexin, as well as hydrophobic substances with features akin to triclosan, such as polychlorinated biphenyls (PCBs) and polycyclic aromatic hydrocarbons (PAHs), highlights their versatility and applicability across a wide range of contaminants.

## Figures and Tables

**Figure 1 molecules-29-02923-f001:**
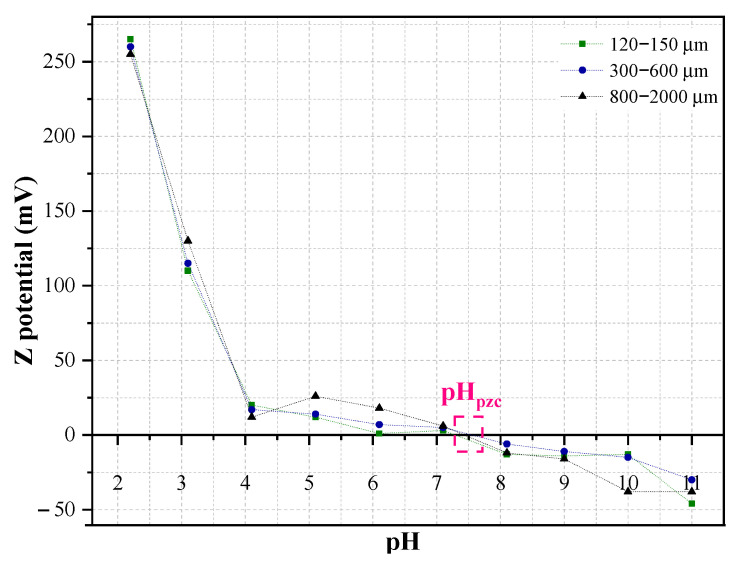
Point of zero charge across different sizes of peanut shell particles.

**Figure 2 molecules-29-02923-f002:**
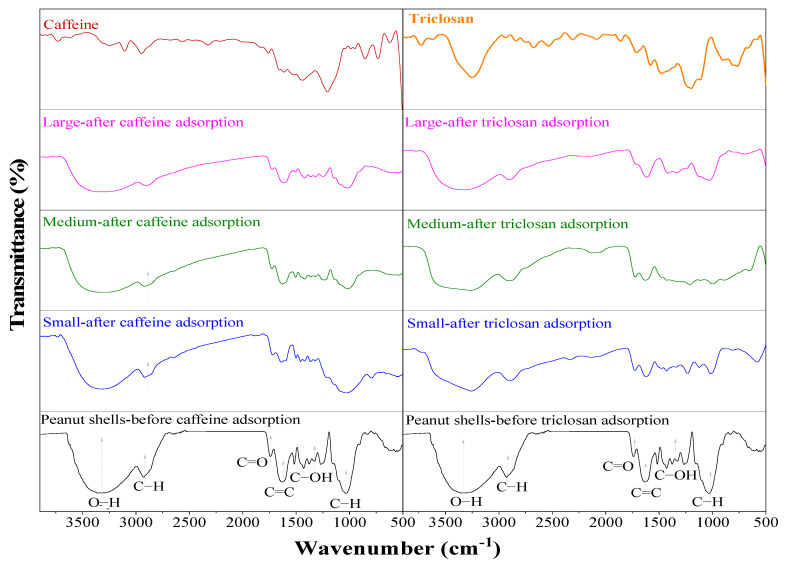
FTIR spectra of peanut shells before and after adsorption of caffeine and triclosan.

**Figure 3 molecules-29-02923-f003:**
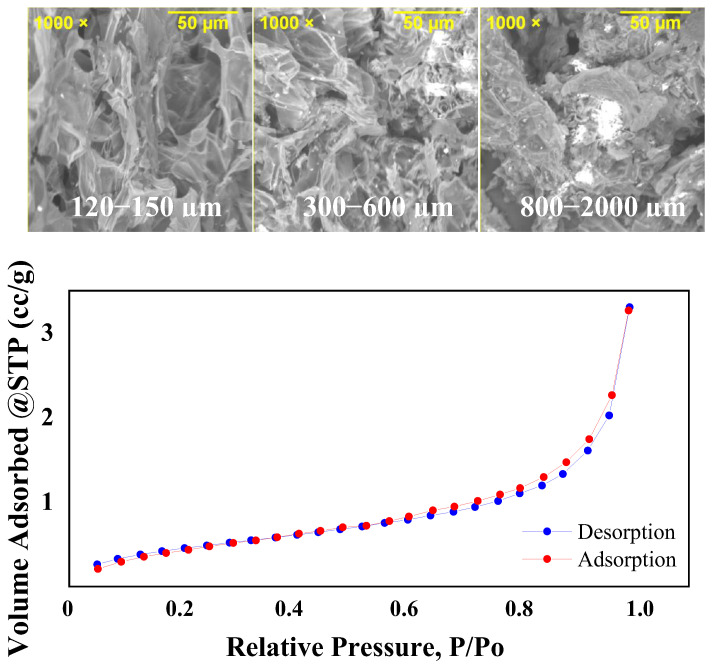
SEM images of small, medium, and large particles and nitrogen adsorption–desorption isotherm.

**Figure 4 molecules-29-02923-f004:**
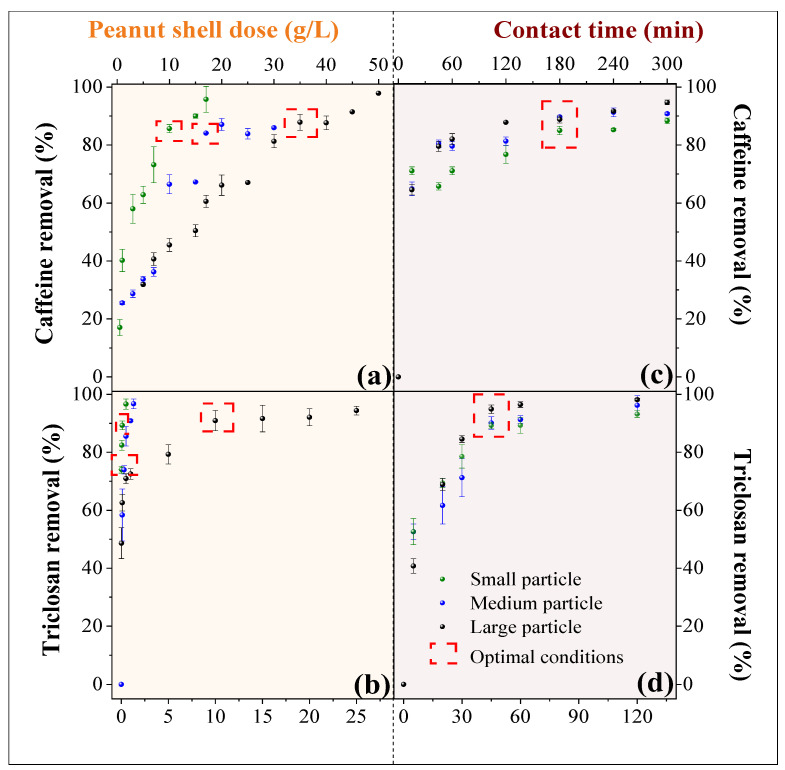
Optimal dose of peanut shells and contact time for the removal of caffeine and triclosan. Green markers: small particles, blue markers: medium particles and red markers: medium particles.

**Figure 5 molecules-29-02923-f005:**
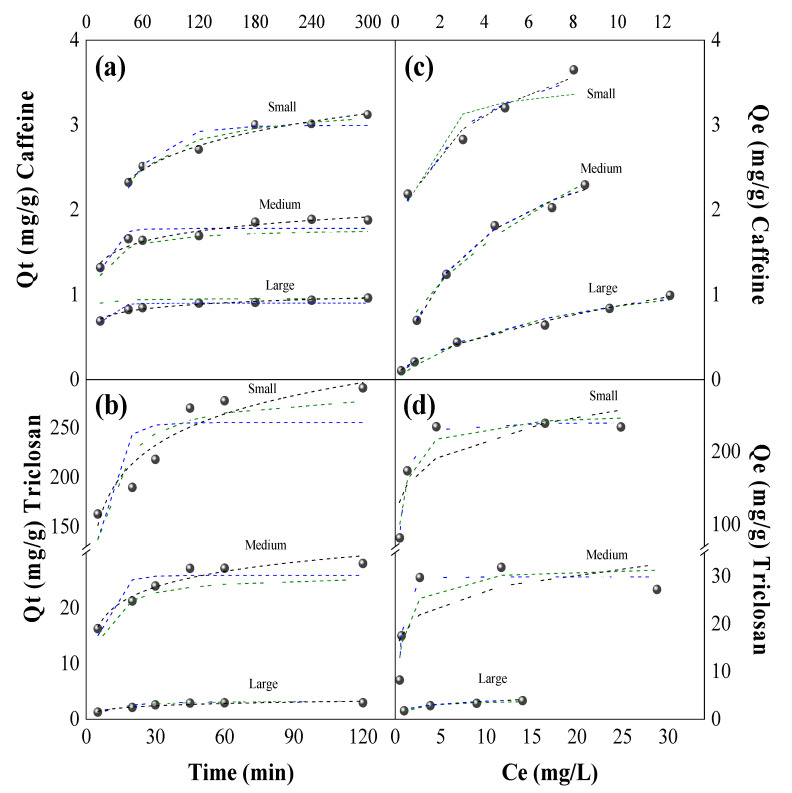
Adsorption kinetics and isotherm models for caffeine and triclosan. Gray points: experimental data, blue line: pseudo-first-order and Sips models, green line: pseudo-second-order and Langmuir models, black line: Elovich and Freundlich model.

**Figure 6 molecules-29-02923-f006:**
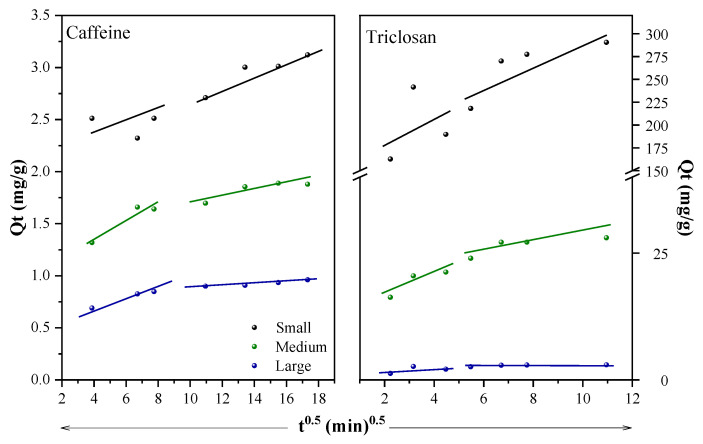
Intraparticle diffusion kinetics for adsorption of caffeine and triclosan.

**Figure 7 molecules-29-02923-f007:**
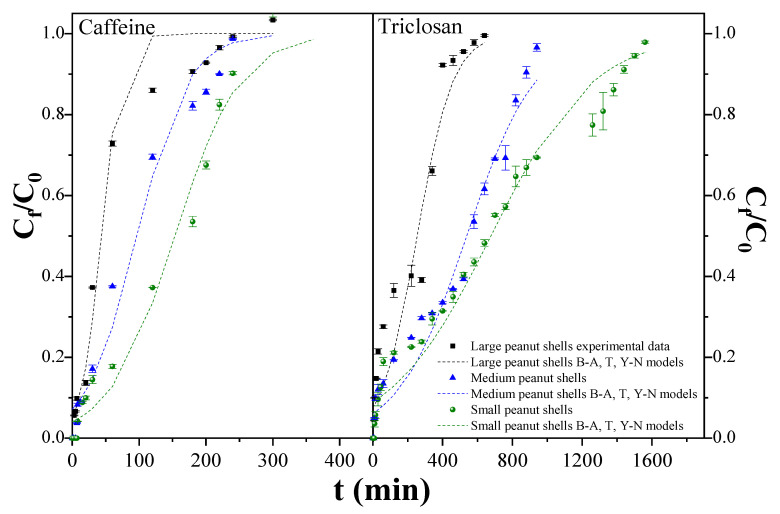
Influences of peanut shell size and type of contaminant on the breakthrough curves. Bed height = 4 cm and hydraulic loading rate = 2 m^3^/m^2^-day. B-A = Bohart–Adams model, T = Thomas model, Y-N = Yoon–Nelson model.

**Table 1 molecules-29-02923-t001:** Chemical composition of peanut shells.

Parameter	Peanut Shells Composition (%)			TGA Analysis
	A [22]	B [23]	Current Study	
Humidity	6.5	-	8.9 ± 0.1	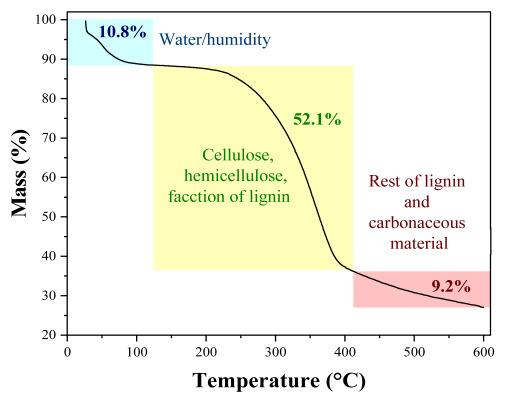
Lignin	30.9	31.6	36.2 ± 4.4	
Cellulose	54.6	34.7	19.4 ± 5.9	
Hemicellulose	14.5	9.0	26.1 ± 1.7	
Extractives	-	13.8	15.2 ± 0.4	
Volatile material	68.8	76.0	74.1 ± 0.4	
Ash	5.5	5.5	3.1 ± 0.1	
Fixed carbon	19.20	-	13.8 ± 0.4	

**Table 2 molecules-29-02923-t002:** Parameters for non-linear adsorption kinetics models for caffeine and triclosan removal.

Model		Caffeine			Triclosan		
		Particle Size (µm)					
		120–150	300–600	800–2000	120–150	300–600	800–2000
	q_exp_ (mg/g)	3.045	1.877	0.938	277.969	27.432	2.914
Pseudo-first-order	Q_e_ (mg/g)	2.993	1.782	0.901	255.350	25.863	2.907
	K_1_ (1/min)	0.031	0.086	0.092	0.154	0.174	0.081
	R^2^	0.868	0.757	0.772	0.509	0.737	0.927
	SSE	0.067	0.059	0.011	67.245	27.087	0.159
	χ^2^	0.004	0.005	0.002	4.805	0.189	0.011
Pseudo-second order	Q_e_ (mg/g)	3.267	1.892	0.951	289.324	28.573	3.270
	K_2_ (g/(g min))	0.016	0.077	1.134	0.001	0.008	0.036
	R^2^	0.964	0.929	0.961	0.766	0.918	0.969
	SSE	0.018	0.017	0.011	31.991	8.455	0.067
	χ^2^	0.001	0.001	0.002	2.280	0.059	0.005
Elovich	α (g/(g min))	2.721	23.043	27.544	235.956	46.928	1.155
	Β mg/g	2.421	5.501	11.924	0.022	0.246	1.685
	R^2^	0.976	0.933	0.966	0.891	0.941	0.919
	SSE	0.008	0.069	9.420 × 10^−4^	87.988	7.110	0.051
	χ^2^	0.003	0.003	0.342 × 10^−4^	3.729	1.515	0.044
Diffusion	K_p1_ (mg/(g min^1/2^))	−0.014	0.091	0.042	8.634	2.116	0.321
	C_1_ (mg/g)	2.534	0.984	0.530	169.595	12.385	0.953
	R^2^	0.866	0.907	0.984	0.583	0.783	0.449
	SSE	0.022	0.007	2.376 × 10^−4^	3.029	3.129	0.683
	K_p2_ (mg/(g min^1/2^))	0.060	0.029	0.010	11.083	0.603	0.059
	C_2_ (mg/g)	2.106	1.420	0.786	178.469	21.913	2.384
	R^2^	0.867	0.7557	0.931	0.669	0.631	0.594
	SSE	0.012	0.006	1.605 × 10^−4^	1.004	3.510	0.039

**Table 3 molecules-29-02923-t003:** Parameters for non-linear isotherm models for caffeine and triclosan removal.

Model		Caffeine			Triclosan		
		Particle Size (µm)					
		120–150	300–600	800–2000	120–150	300–600	800–2000
	q_exp_ (mg/g)	3.649	2.293	0.993	238.906	31.849	3.882
Langmuir	q_max_ (mg/g)	3.522	3.147	1.490	253.559	32.049	4.041
	K_L_ (L/mg)	2.619	0.289	0.139	1.317	1.396	0.707
	R^2^	0.838	0.993	0.977	0.947	0.838	0.959
	SSE	0.186	0.012	0.014	9.627	6.338	0.100
	χ^2^	0.016	0.001	0.005	0.995	0.548	0.008
Freundlich	K_F_ ((mg/g)/ (mg/L))^n^	2.370	0.804	0.225	145.638	18.328	1.846
	1/n	0.196	0.492	0.582	0.176	0.168	0.280
	R^2^	0.976	0.980	0.995	0.719	0.561	0.985
	SSE	0.028	0.033	0.003	510.919	17.161	0.036
	χ^2^	0.002	0.004	0.001	5.288	1.490	0.003
Sips	q_max_ (mg/g)	6.899	3.273	1.978	238.975	29.647	7.376
	K_S_ (L/mg)	0.141	0.264	0.072	1.359	1.585	0.083
	1/n	0.325	0.959	0.802	1.737	3.645	0.456
	R^2^	0.958	0.993	0.987	0.997	0.973	0.989
	SSE	0.049	0.012	0.008	5.335	1.072	0.028
	χ^2^	0.004	0.001	0.003	0.055	0.094	0.002

q_exp_ = equilibrium adsorption capacity in experimental testing.

**Table 4 molecules-29-02923-t004:** Principal outcomes from the fixed-bed column adsorption tests.

Parameter		Contaminants					
		Particle Size (µm)					
		Caffeine			Triclosan		
		120–150	300–600	800–2000	120–150	300–600	800–2000
Mass (g)		0.80	1.40	1.43	0.80	1.40	1.43
V_c_ (L)		3.14 × 10^−3^					
EBTC (day)		0.44					
FBU (%)		77.41	50.81	37.42	47.86	36.52	23.76
h_MTZ_ (cm)		0.90	1.97	2.50	1.14	2.54	3.05
C/C_0_ = 0.1	t_b_ (min)	20.20	12.20	7.00	30.20	15.20	8.20
	V_b_ (mL)	10.10	6.10	3.50	15.10	7.60	4.10
	q_b_ (mg/g)	0.26	0.16	0.07	0.80	0.27	0.12
C/C_0_ = 0.9	t_s_ (min)	240.20	220.20	180.2	1440.20	880.20	400.20
	q_s_ (mg/g)	0.33	0.31	0.19	1.12	0.74	0.49

**Table 5 molecules-29-02923-t005:** Parameters for caffeine and triclosan adsorption according to the Bohart–Adams, Thomas, and Yoon–Nelson models.

Model		Contaminants					
		Caffeine			Triclosan		
		Particle Size (µm)					
		120–150	300–600	800–2000	120–150	300–600	800–2000
Bohart–Adams	K_AB_ (L/(min-mg))	6.361 × 10^−4^	8.184 × 10^−4^	2.072 × 10^−3^	7.655 × 10^−5^	1.134 × 10^−4^	2.193 × 10^−4^
	N_0_ (mg/L)	700.941	442.586	197.856	4293.136	3411.516	1612.556
Thomas	q_0_ (mg/g)	0.716	0.453	0.203	143.438	2.511	1.187
	k_TH_ (mg/min)	1.919	2.475	6.335	0.006	0.475	0.920
Yoon–Nelson	k_YN_ (min^−1^)	0.020	0.026	0.068	0.003	0.005	0.010
	Τ (min)	153.751	97.152	43.427	677.226	538.153	254.545

## Data Availability

Data is contained within the article.

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
