# Peer review of "Characterization and Performance of Peanut Shells in Caffeine and Triclosan Removal in Batch and Fixed-Bed Column Tests"

_molecules, 2024, doi:10.3390/molecules29122923_

Round 1

Reviewer 1 Report

Comments and Suggestions for Authors

Comments regarding the manuscript entitled "Characterization and performance of peanut shells for caffeine and triclosan removal in batch and fixed-bed column tests" (Ref.: Manuscript ID: molecules-3018128).

The paper's subject is interesting because of its implications on health and the environment. In addition, the manuscript is easy to follow. However, some criticisms can be made of the present version of the manuscript. My principal comments are as follows:

1) Line 76: The full meaning of an abbreviation should be defined at first mention in the text. Please define TCS.

2) Figure 2: Provide the SEM images of small, medium, and large peanut shell particles after saturation with caffeine and triclosan and compare them with the spectra of unsaturated particles.

3) Lines 219-224: To have caffeine removal efficiencies of 84-88%, very high concentrations of peanut shell particles are required (10-35 g/L). Hence, the regeneration of the biosorbent is of paramount importance to avoid secondary pollution problems. Provide data on caffeine desorption and peanut shell regeneration. 

4) Lines 219-224: Provide caffeine and triclosan biosorption capacities of the small, medium, and large peanut shell particles.

5) Lines 231-232: Do substances leached from peanut shells interfere with the spectrophotometric determination of caffeine and triclosan?

6) Lines 222-225 and 237-241: Are there statistically significant differences between the caffeine and triclosan removal efficiencies of the small, medium, and large peanut shell particles? Analyze and discuss the results based on statistical analysis to determine if significant differences exist.

7) Tables 2 and 3: Please provide the experimental caffeine and triclosan biosorption capacities at equilibrium.

8) Table 2 and Figures 4a and 4b: The biosorption capacities of caffeine and triclosan reported in Table 2 (caffeine: 3.27 - 289.3 mg/g (pseudo-second-order model); triclosan: 0.951 - 3.267 mg/g (pseudo-second-order model)) do not coincide with those of Figures 4a (caffeine: < 4 mg/g) and 4b (triclosan: > 250 mg/g). Please check throughout the manuscript. 

9) Table 3 and Figures 4c and 4d: The equilibrium biosorption capacities of caffeine and triclosan reported in Table 3 (caffeine: 7.376 – 238.97 mg/g (Sips model); triclosan: 1.978 - 6.899 mg/g (Sips model)) do not coincide with those of Figures 4c (caffeine: < 4 mg/g) and 4d (triclosan: > 200 mg/g). Please check throughout the manuscript.

10) Figure 4: There is no red line (see Figure footnote).

11) Section 2.4: No information is provided on the characterization of the packed bed, e.g., porosity, bulk density, etc.

12) Solution pH is among the most critical factors affecting the biosorption of adsorbates. However, the authors did not determine the effect of pH on caffeine and triclosan biosorption by peanut shell. Why? 

13) Lines 496-497: Hydraulic loading rates of 2 and 4 m3/m2×d were tested in the work (no results are presented for a hydraulic load of 4 m3/m2×d). The authors stated verbatim that these hydraulic loads are consistent with those used with commercial adsorbents such as granular activated carbon and cited the reference of Sandoval et al. (2011). However, Sandoval et al. reported a hydraulic loading rate of 2.8 L/m2×s = 241.92 m3/m2×d, which is approximately 121 times higher than 2 m3/m2×d. 

14) Lines 536-537: The authors stated verbatim that the intraparticle diffusion model (equation 5) was applied to obtain information about the mechanism acting in the caffeine/triclosan adsorption process. However, no results are provided in the manuscript.

15) Equation 13: Define h.

Comments on the Quality of English Language

In general the manuscript is well written, only a few typographycal errors were detected.

Author Response

Dear Reviewer,

We would like to thank you for the comments and suggestions for the manuscript entitled: “Characterization and Performance of Peanut Shells for Caffeine and Triclosan Removal in Batch and Fixed-Bed Column Tests”. Following the suggestions received, we have modified the manuscript, added the suggested references, and carefully revised the complete document. As a result, we believe that our manuscript has improved substantially, and conveys in a better manner some key points of our research.

Best regards,

Cristina Almeida

Reviewer 2 Report

Comments and Suggestions for Authors

In this manuscript the possibility of using peanut shells for water purification (specifically the removal of caffeine and triclosan) is discussed. Peanut shells were characterized using scanning electron microscopy and infrared spectroscopy (ATR mode) in the context of their chemical composition and the ability to absorb water pollutants of different chemical properties. This paper describes a “green” (and residual) material for water purification – this approach is in close agreement with contemporary trends. 

The Authors carefully analyzed several factors influencing the absorption processes on peanut shell material of different particle sizes. Caffeine and triclosan removal data were fitted to different  models, separately for different particle sizes.

Removed solutes differ a lot in terms of their physico-chemical properties – they were selected in such a way that this methodology may  be applied to a broad range of structurally diverse compounds.

This is definitely not the first example of residual agricultural material being used for this purpose, but as it contains a very detailed analysis of parameters that may contribute to the quality of the purification process, in my opinion it provides the reader with some interesting knowledge and therefore, after some minor improvements, it deserves consideration.

In my opinion some proposals of further applications of this methodology would be an asset (to what solutes?  Some examples, maybe?). I would also very much appreciate some suggestion on used peanut shells disposal – I assume that they are not supposed to be re-used, so do the Authors suggest e.g. combustion or is there any other idea?
The references are appropriate.
I would much appreciate expanding section 3.3.2. – calibration curves (figures or just equations, R2 etc.) and spectroscopic data for caffeine and triclosan concentration measurements should be added.

Minor comments:

1. Could the Authors please check the equations once again, especially equation 4

2. Figure 4 is difficult to read.

Author Response

(The authors gave the same response as above.)

Round 2

Reviewer 1 Report

Comments and Suggestions for Authors

The authors considered my comments and suggestions and have updated their manuscript. I have only three additional minor comments:

1)        A typographical error is in line 338 (monolayer formation o.0f the Langmuir).

2)        Check the number of equations. There is no equation 14.

3)        Equations 17-19: Some squares can be seen in equations 17-19 for the indexes of the Greek capital letters S, but no information is provided.

Author Response

We sincerely appreciate the reviewer's collaboration in improving the document. Below, we address the observations, which have been highlighted in the main document.

The authors considered my comments and suggestions and have updated their manuscript. I have only three additional minor comments:

  • A typographical error is in line 338 (monolayer formation 0f the Langmuir).

The typographical error in line 338 was corrected.

  • Check the number of equations. There is no equation 14.

We have thoroughly reviewed the numbering of the equations and have made the necessary corrections. The document now correctly includes all three equations.

3)        Equations 17-19: Some squares can be seen in equations 17-19 for the indexes of the Greek capital letters S, but no information is provided.

The Greek letter S in equations 17-19 should not contain squares. This has been modified to avoid any confusion.
